

# An integrated platform for intuitive mathematical programming modeling using LaTeX

Charalampos P. Triantafyllidis[1,2] and Lazaros G. Papageorgiou[1]

[1] Centre for Process Systems Engineering, Department of Chemical Engineering, University College London, London, United Kingdom
[2] Smith School of Enterprise and the Environment, University of Oxford, Oxford, United Kingdom

## ABSTRACT

This paper presents a novel prototype platform that uses the same LaTeX mark-up language, commonly used to typeset mathematical content, as an input language for modeling optimization problems of various classes. The platform converts the LaTeX model into a formal Algebraic Modeling Language (AML) representation based on Pyomo through a parsing engine written in Python and solves by either via NEOS server or locally installed solvers, using a friendly Graphical User Interface (GUI). The distinct advantages of our approach can be summarized in (i) simplification and speed-up of the model design and development process (ii) non-commercial character (iii) cross-platform support (iv) easier typo and logic error detection in the description of the models and (v) minimization of working knowledge of programming and AMLs to perform mathematical programming modeling. Overall, this is a presentation of a complete workable scheme on using LaTeX for mathematical programming modeling which assists in furthering our ability to reproduce and replicate scientific work.

## INTRODUCTION

Mathematical modeling constitutes a rigorous way of inexpensively simulating complex systems' behavior in order to gain further understanding about the underlying mechanisms and trade-offs. By exploiting mathematical modeling techniques, one may manipulate the system under analysis so as to guarantee its optimal and robust operation.

The dominant computing tool to assist in modeling is the Algebraic Modeling Languages (AMLs) (*Kallrath, 2004*). AMLs have been very successful in enabling a transparent development of different types of models, easily distributable among peers and described with clarity, effectiveness and precision. Software suites such as AIMMS (*Bisschop & Roelofs, 2011*), GAMS IDE (*McCarl et al., 2013*), JuMP (*Dunning, Huchette & Lubin, 2017*) as the modeling library in Julia (*Lubin & Dunning, 2015*), Pyomo (http://www.Pyomo.org/) (*Hart et al., 2017*; *Hart, Watson & Woodruff, 2011*) for modeling in Python (https://www.python.org/), (*Rossum, 1995*) and AMPL (*Fourer, Gay*

Corresponding author
Lazaros G. Papageorgiou,
l.papageorgiou@ucl.ac.uk

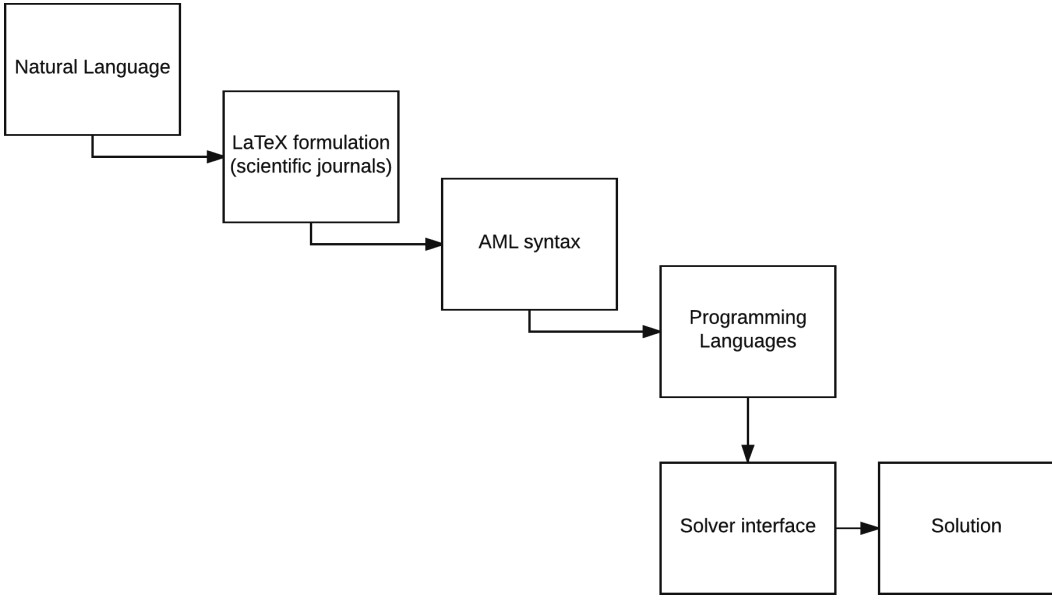

**Figure 1** The levels of abstraction in modeling; from natural language to extracting the optimal solution via computational resources.

*& Kernighan, 1993*) are the most popular and widely used in both academia and industry. AMLs usually incorporate the following features:

- a strict and specific syntax for the mathematical notation to describe the models;
- solver interfaces, the bridge between mathematics and what the solver can *understand* in terms of structural demands;
- a series of available optimization solvers for as many classes of problems as supported (LP, MILP, MINLP etc.) with the associated functional interfaces implemented;
- explicit data file formats and implementation of the respective import/export mechanisms.

AMLs provide a level of abstraction, which is higher than the direct approach of generating a model using a programming language. The different levels in the design process of a model are depicted in Fig. 1. Extending an AML (or even the entire modeling design process) can be done in the following two ways: we can either simplify the present framework (*vertical abstraction*) or extend the embedded functionality (*horizontal abstraction*) (*Jackson, 2012*). The layers of abstraction between the conception and the semantics of a mathematical model and its computational implementation may not necessarily be *thin*. This means that while eventually the aim of the presented platform has the same purpose as an AML that is to generate and solve models, simplification of the required syntax to describe the model is associated with higher complexity. Thus, in order to relax the syntactical requirements, we have to be able to process the same model with limited information (for instance, we do not declare index sets and parameters in the platform). This limited declaration of model components elevates the amount of processing that the platform has to conduct in order to provide equivalent formulations of the input.

A systems approach, MOSAIC (*Erik et al., 2017*), has been developed based on a MathML representation using LaTeX extracts, which has been applied mainly to chemical engineering models. Both frameworks can be facilitated online, with the proposed framework built on Django while MOSAIC on Java. It can be noted that our platform can also be run off-line (locally). A key difference between the two is that in the proposed framework the user does not explicitly define indices, parameters and dynamic sets as those are identified automatically from the platform, by filtering them out from the variable list given at the bottom of the input .tex model. In the proposed platform the user can capture the entire optimization model in a single .tex file and use this directly as an input to the platform as opposed of using LaTeX extracts for generating equations in MOSAIC. Similarly though, both platforms are framing the use of LaTeX built-in commands for the specific environment to better capture errors and provide more consistency. Finally, the proposed platform exports the generated optimization model in Pyomo whereas the ability to export in many other formats is given in the MOSAIC environment.

Our work expands upon two axes: (i) the programming paradigm introduced by Donald E. Knuth (*Knuth, 1984*) on *Literate Programming* and (ii) the notions of *reproducible and replicable research*, the fundamental basis of scientific analysis. Literate Programming focuses on generating programs based on logical flow and thinking rather than being limited by the imposing syntactical constraints of a programming language. In essence, we employ a simple mark-up language, LaTeX, to describe a problem (mathematical programming model) and then in turn produce compilable code (Pyomo abstract model) which can be used outside of the presented prototype platform's framework. Reproducibility and the ability to replicate scientific analysis is crucial and challenging to achieve. As software tools become the vessel to unravel the computational complexity of decision-making, developing open-source software is not necessarily sufficient; the ability for the averagely versed developer to reproduce and replicate scientific work is very important to effectively deliver impact (*Leek & Peng, 2015*; *Nature Methods Editorial Board, 2014*). To quote the COIN-OR Foundation (https://www.coin-or.org/), *Science evolves when previous results can be easily replicated.*

In the endeavor of simplifying the syntactical requirements imposed by AMLs we have developed a prototype platform. This new framework is materializing a level of modeling design that is higher than the AMLs in terms of *vertical abstraction*. It therefore strengthens the ability to reproduce and replicate optimization models across literature for further analysis by reducing the demands in working knowledge of AMLs or coding. The key capability is that it parses LaTeX formulations of mathematical programs (optimization problems) directly into Pyomo abstract models. The framework then combines the produced abstract model with data provided in the AMPL *.dat* format (containing parameters and sets) to produce a concrete model. This capability is provided through a graphical interface which accepts LaTeX input and AMPL data files, parses a Pyomo model, solves with a selected solver (CPLEX, GLPK, or the NEOS server), and returns the optimal solution if feasible, as the output. The aim is not to substitute but to establish a link between those using a higher level of abstraction. Therefore, the platform does not eliminate the use of an AML or the advantages emanating from it.

This is a complete prototype workable scheme to address how LaTeX could be used as an input language to perform mathematical programming modeling, and currently supports Linear Programming (LP), Mixed-Integer Linear Programming (MILP) as well as Mixed-Integer Quadratic Programming (MIQP) formulations. Linear Optimization (*Bertsimas & Tsitsiklis, 1997*; *Williams, 1999*) has proven to be an invaluable tool for decision support. The corpus of models invented for linear optimization over the past decades and for a multitude of domains has been consistently increasing. It can be easily demonstrated with examples in Machine Learning, Operations Research and Management Science, Physics, Information Security, Environmental Modeling and Systems Biology among many others (*Yang et al., 2016*; *Tanveer, 2015*; *Silva et al., 2016*; *Sitek & Wikarek, 2015*; *Liu & Papageorgiou, 2018*; *Triantafyllidis et al., 2018*; *Cohen et al., 2017*; *Romeijn et al., 2006*; *Mitsos et al., 2009*; *Melas et al., 2013*; *Kratica, Dugošija & Savić, 2014*; *Mouha et al., 2012*).

This paper is organized as follows: in 'Functionality', we describe the current functionality supported by the platform at this prototype stage. In 'Parser - Execution Engine', we present the implementation details of the parser. 'An illustrative parsing example' provides a description of an illustrative example. A discussion follows in 'Discussion'. Some concluding remarks are drawn in 'Conclusion'. Examples of optimization models that were reproduced from scientific papers as well as their corresponding LaTeX formulations and Pyomo models can be found in the Supplemental Information 1.

## FUNCTIONALITY

The set of rules that are admissible to formulate models in this platform are formal LaTeX commands and they do not represent in-house modifications. We assume that the model will be in the typical format that optimization programs commonly appear in scientific journals. Therefore, the model must contain the following three main parts and with respect to the correct order as well:

1. the objective function to be optimized (either maximized or minimized);
2. the (sets of) constraints, or else the relationships between the decision variables and coefficients, right-hand side (RHS);
3. the decision variables and their domain space.

We used the programming environment of Python coupled with its modeling library, namely Pyomo. Similar approaches in terms of software selection have been presented for Differential and Algebraic Equations (DAE) modeling and optimization in (*Nicholson et al., 2018*; *Nikolić, 2016*). By combining Python and Pyomo we have the ability to transform a simplified representation of a mathematical model initially written in LaTeX into a formal AML formulation and eventually optimize it. In other words, the platform *reads* LaTeX code and then *writes* Pyomo abstract models or *the code generates code*. The resulting *.py* file is usable outside of the platform's frame, thus not making the binding and usage of these two necessary after conversion. The main components that we employed for this purpose are the following:

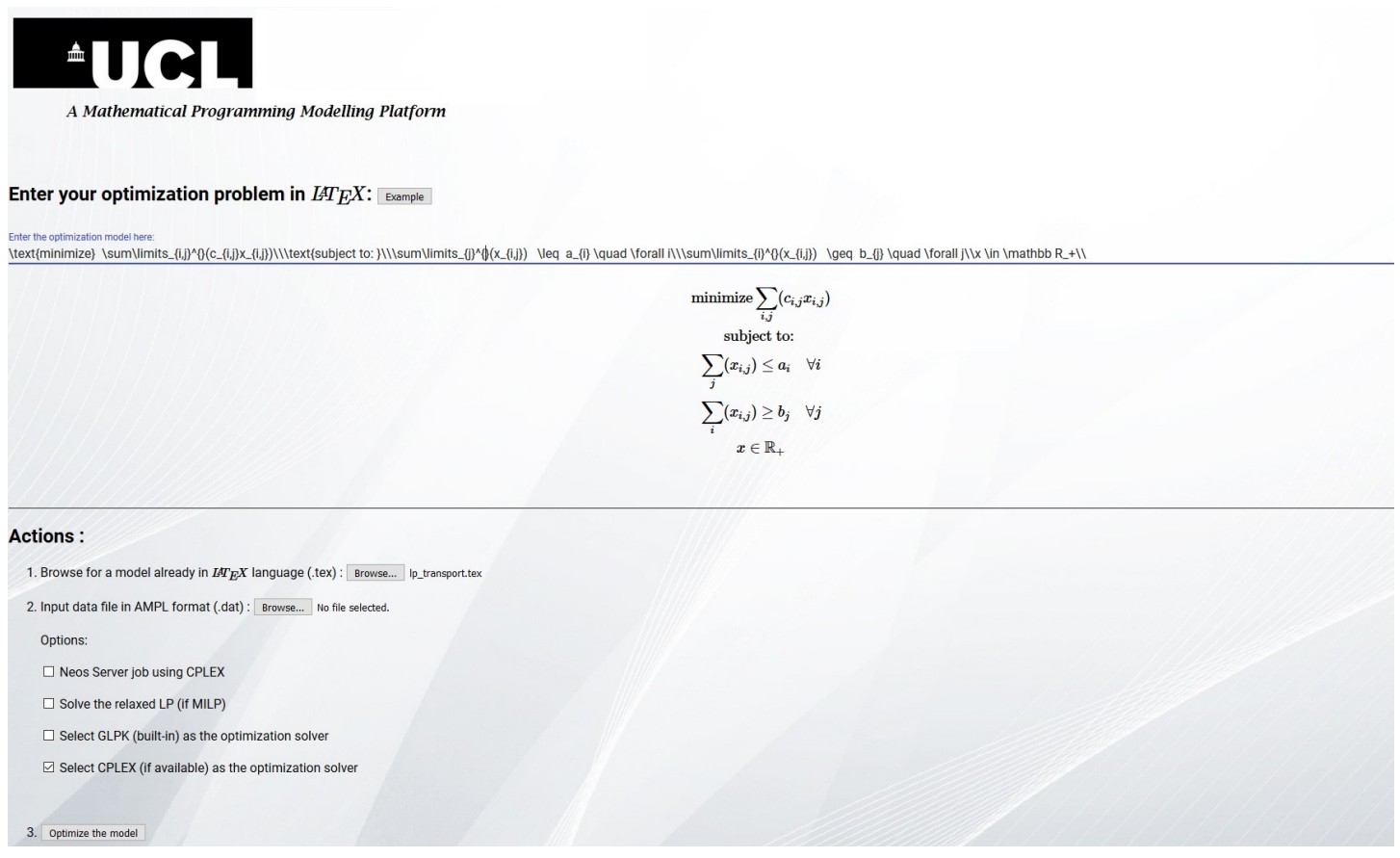

**Figure 2** **The simplified Graphical User Interface (GUI).** The GUI contains the basic but fundamental options to use the platform, such as model input, solver selection and solution extraction.

- Front-end: HTML, JavaScript, MathJax (https://www.mathjax.org/) and Google Polymer (https://www.polymer-project.org/);
- Back-end: Python with Django (https://www.djangoproject.com/) and Pyomo.

In order to increase the effectiveness and user-friendliness of the platform, a Graphical-User Interface (GUI) based on HTML, JavaScript (front-end) and Django as the web-framework (back-end) has been implemented, as shown in Fig. 2. The user-input is facilitated mainly via Polymer objects (https://www.polymer-project.org/). As the main feature of the platform is to allow modeling in LaTeX language, we used MathJax as the rendering engine. In this way, the user can see the compiled version of the input model. All of these components form a single suite that works across different computational environments. The front-end is plain but incorporates the necessary functionality for input and output, as well as some solver options. The role of the back-end is to establish the communication between the GUI and the parser with the functions therein. In this way the inputs are being processed inside Python in the background, and the user simply witnesses a seamless working environment without having to understand the *black-box* parser in detail.

The main components of the GUI are:

- *Abstract model input*: The input of the LaTeX model, either directly inside the Polymer input text-box or via file upload (a *.tex* containing the required source LaTeX code)
- *Data files*: The input of the data set which follows the abstract definition of the model via uploading the AMPL-format (*.dat*) data file
- *Solver options*: An array of solver - related options such as:
  1. NEOS server job using CPLEX
  2. Solve the relaxed LP (if MILP)
  3. Select GPLK (built-in) as the optimization solver
  4. Select CPLEX (if available) as the optimization solver (currently set to default)

The following is an example of a LaTeX formulated optimization problem which is ready to use with the platform; the well-known Traveling Salesman Problem (TSP) (*Applegate et al., 2007*):

$$\texttt{minimize} \qquad \sum_{i,j:i\neq j}(d_{i,j}x_{i,j})$$

$$\texttt{subject to}:$$

$$\sum_{j:i\neq j}(x_{i,j})=1 \qquad \forall i$$

$$\sum_{i:i\neq j}(x_{i,j})=1 \qquad \forall j$$

$$u_i-u_j+nx_{i,j}\leq n-1 \qquad \forall i\geq 2, j\leq |j|-1, i\neq j$$

$$u\in\mathbb{Z}, x\in\{0,1\}$$

and the raw LaTeX code used to generate this was:

```
\text{minimize}   \sum\limits_{i,j: i \neq j}^{} (d_{i,j}x_{i,j})\\
\text{subject to: }\\
\sum\limits_{j: i \neq j}^{} (x_{i,j}) =  1 \quad \quad \forall i\\
\sum\limits_{i: i \neq j}^{} (x_{i,j}) =  1 \quad \quad \forall j\\
u_{i} - u_{j} + nx_{i,j} \leq n - 1   \quad \quad \forall i \geq 2, j \leq |j|-1, i
    \neq j\\
u \in \mathbb Z, x \in \{0,1\}\\
```

which is the input for the platform. The user can either input this code directly inside the Google polymer text box or via a *pre-made .tex* file which can be uploaded in the corresponding field of the GUI. Either way, the MathJax Engine then renders LaTeX appropriately so the user can see the resulting compiled model live. Subject to syntax-errors, the MathJax engine might or might not render the model eventually, as naturally expected. Empty lines or spaces do not play a role, as well as commented-out lines using the standard notation (the percentage symbol %). The model file always begins with the objective function sense, the function itself, and then the sets of constraints follow, with the variables and their respective type at the end of the file.

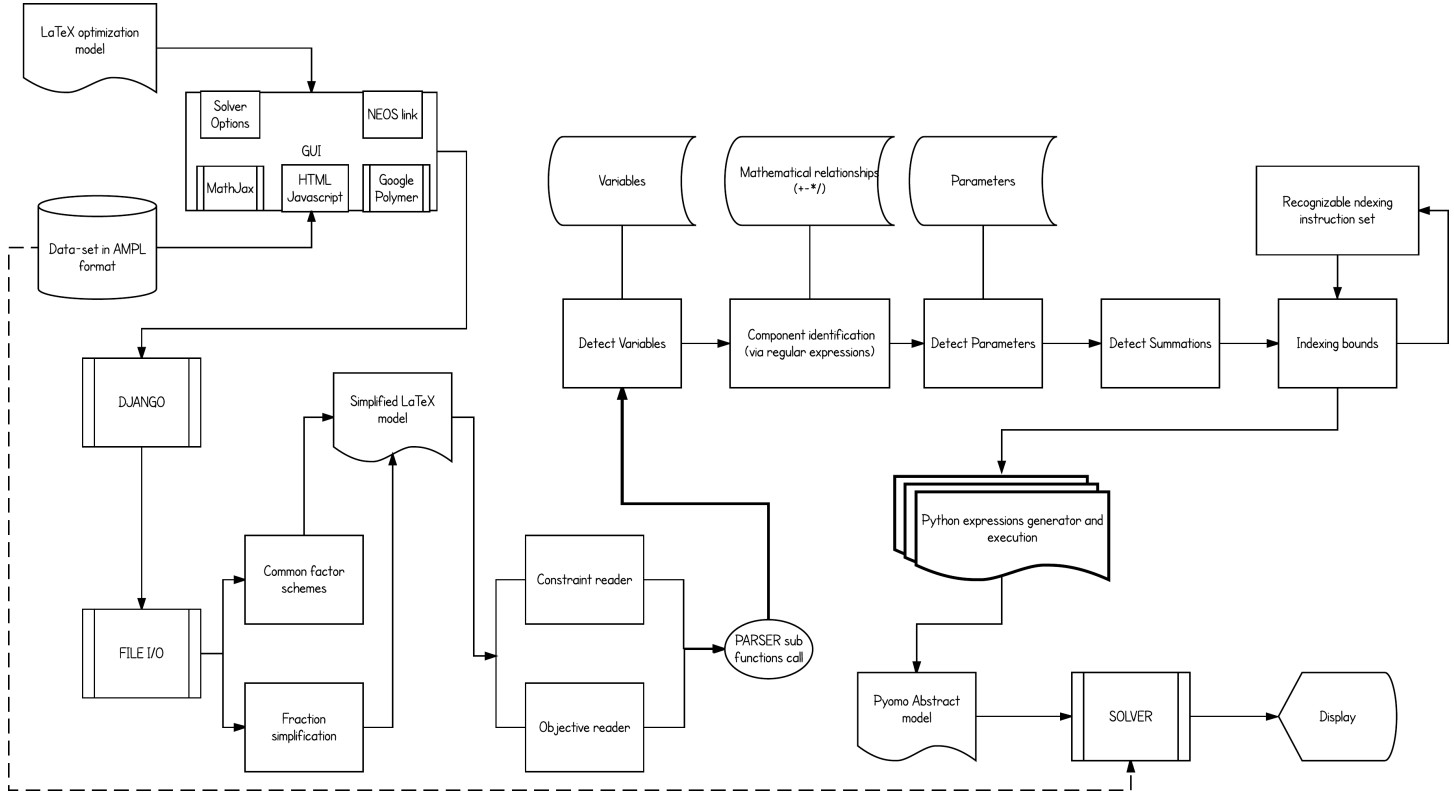

**Figure 3** **The overall flow of the implementation.** From user input to solving the optimization problem or simply exporting the equivalent Pyomo model file.

## PARSER—EXECUTION ENGINE

As *parser* we define the part of the code (a collection of Python functions) in the back-end side of the platform which is responsible for translating the model written in LaTeX to Pyomo, the modeling component of the Python programming language. In order to effectively translate the user model input from LaTeX, we need an array of programming functions to carry out the conversion consistently since preserving the equivalence of the two is implied. The aim of the implementation is to provide minimum loss of generality in the ability to express mathematical notation for different modeling needs.

A detailed description of the implemented scheme is given in Fig. 3. A modular design of different functions implemented in Python and the established communication of those (exchanging input and output-processed data) form the basic implementation concept. This type of design allows the developers to add functionality in a more clear and effective way. For instance, to upgrade the parser and support Mixed Integer Quadratic Programming (MIQP) problems, an update only to the parsing function assigned to convert the optimization objective function is required.

Once the *.tex* model file and the *.dat* AMPL formatted data file are given, the platform then starts processing the model. The conversion starts by reading the variables of the model and their respective types, and then follows with component identification

(locating the occurrence of the variables in each constraint) and their inter-relationships (multiplication, division, summation etc.). Additionally, any summation and constraint conditional indexing schemes will be processed separately. Constraint-by-constraint the parser gradually builds the *.py* Pyomo abstract model file. It then merges through Pyomo the model with its data set and calls the selected solver for optimization.

## Pre-processing

A significant amount of pre-processing takes place prior of parsing. The minimum and essential is to first tidy up the input; that is, clear empty lines and spaces, as well as reserved (by the platform) keywords that the user can include but do not play any role in functional parsing (such as the \\*quad* command). The platform also supports the use of Greek letters. For instance, if a parameter is declared as $\alpha$ the platform identifies the symbol, removes the backslash and expects to find *alpha* in the data-file. This takes place also in the pre-processing stage.

The user can also opt-out selectively the constraints by putting regular comments in LaTeX, with the insertion of the percentage symbol (%) in the beginning of each expression. Once done, we attempt to simplify some types of mathematical expressions in order to be able to better process them later on. More specifically, we have two main functions that handle fractions and common factor (distributive expressions) simplifications. For example:

$\dfrac{A_i B_j}{D_i}$ is then converted to: $(A_i B_j)/D_i$

and

$\beta(\alpha + 1)$ is converted as expected to: $\beta\alpha + \beta$

when handling fractions, the user can employ the *frac* environment to generate them; the parser in the background always though processes the analytic form (the same applies with the distributive form of multiplications), no matter if the initial input was done using the *frac* environment.

This keeps the basic component identification functions intact, since their input is transformed first to the acceptable analytical format. Instead of transforming the parsing functions, we transform the input in the acceptable format. However, the user does not lose either functionality or flexibility, as this takes place in the background. To put it simply, either the user inputs the analytic form of an expression or the compact, the parser is still able to function correctly.

To frame the capabilities of the parser, we will now describe how the user can define optimization models in the platform with a given example and the successful parsing to Pyomo. The parser first attempts to split the model into its three major distinct parts:

- the objective function
- the sets of constraints
- the types of the variables defined

These three parts are in a way independent but interconnected as well.

## Processing variables

The parser first attempts to read the variables and their respective domain space (type). The platform is case sensitive since it is based on Pyomo. The processing is done using *string* manipulation functions, therefore the use of *regular expressions* in Python was essential and effective.

Reasonably, the focus was on consistency and reliability, rather computational performance mainly due to the lightweight workload of the processing demands in general. In order to do that, the parser uses *keywords* as *identifiers* while scanning from the top to the bottom of the manually curated *.tex* file which contains the abstract model in LaTeX. For the three respective different parts mentioned earlier, the corresponding identifiers are:

1. Objective function: {*minimize*, *maximize*}
2. Sets of constraints: {\leq, \geq, =}
3. Variables and their types: {\mathbb , {0, 1}}

This helps separate the processing into sections. Each section is analyzed and passes the information in Pyomo syntax in the *.py* output model file. Variable types can appear in the following way:

- `\in \mathbb R`
  for Real numbers ($\in \mathbb{R}$)
- `\in \mathbb R_+`
  for non-negative Real numbers ($\in \mathbb{R}_+$)
- `\in \mathbb R_{*}^{+}`
  for positive Real numbers ($\in \mathbb{R}_*^+$)
- `\in \{0,1\}`
  for binary variables ($\in \{0, 1\}$)
- `\in \mathbb Z`
  for integers ($\in \mathbb{Z}$)
- `\in \mathbb Z_+`
  for non-negative integers ($\in \mathbb{Z}_+$)
- `\in \mathbb Z_{*}^{+}`
  for positive integers ($\in \mathbb{Z}_*^+$)

In order to avoid confusion between lowercase and uppercase, the identifiers are converted to uppercase prior of comparison. Upon locating these keywords, the parser separates the processing and starts calling the corresponding functions. Once the variables and their types are processed (expected to be found at the bottom of the mathematical definition of the model), the parser then creates a list of strings for the names of the variables. This is one of the crucial structures of the parser and utilized alongside the entire run-time of the conversion process. A list of the same length, which holds the types of each respective variable, is also created. The platform in general uses Python lists to store information about variables, index sets, parameters, scalars etc.

## Decomposing constraints and objective function expressions

Our approach for understanding the inter-mathematical relationships between the variables and the parameters relied on exploiting the fundamental characteristics of Linear Programming:

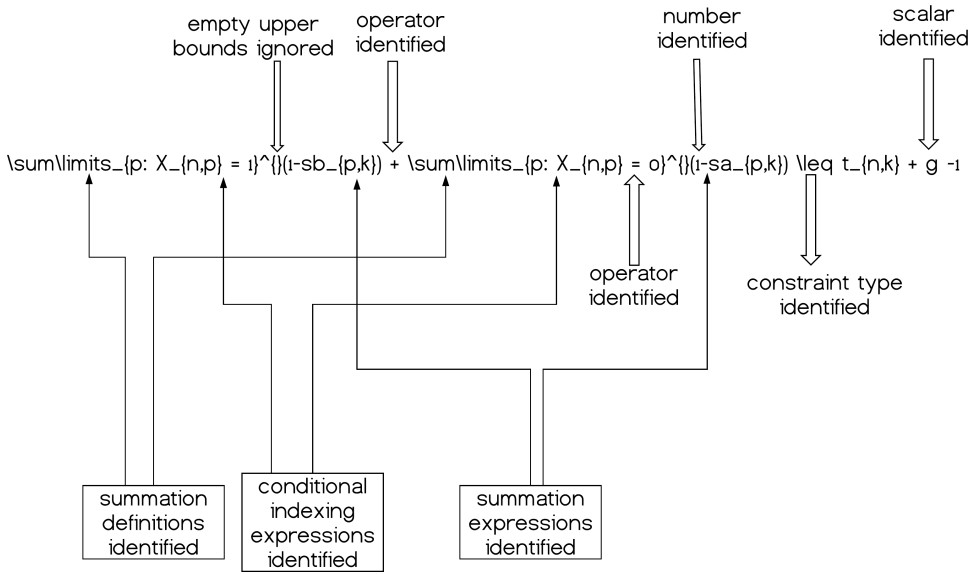

**Figure 4** A simple constraint having its components (partially) decomposed and therefore identified; summations, operators, scalars and numerical quantities.

- Proportionality
- Additivity
- Divisibility

These mathematical relationships can help us understand the structure of the expressions and how to *decompose* them. By *decomposition* we define the fragmentation of each mathematical expression at each line of the .tex input model file into the corresponding variables, parameters, summations etc. so as we can process the given information accordingly. A simple graphical example is given in Fig. 4.

The decomposition with the regular expressions is naturally done via the strings of the possible operators found, that is: addition, subtraction, division $(+, -, /)$, since the asterisk to denote multiplication ($*$ or $\cdot$) is usually omitted in the way we describe the mathematical expressions (e.g., we write $ax$ to describe coefficient $a$ being multiplied by variable $x$). In some cases however it is imperative to use the asterisk to decompose a multiplication. For example, say $Ds$ is a parameter and $s$ is also a variable in the same model. There is no possible way to tell whether the expression $Ds$ actually means $D^*s$ or if it is about a new parameter altogether, since the parameters are not explicitly defined in the model definition (as in AMLs). Adding to that the fact that for the scalars there is no associated underscore character to identify the parameter as those are not associated with index sets, the task is even more challenging. Therefore, we should write $D^*s$ if $D$ is a scalar. As for parameters with index sets, for example $Ds_i s_i$ causes no confusion for the parser because the decomposition based on the underscore character clearly reveals two separate components. In this way, the platform also identifies new parameters. This means that since we know, for instance, that $s$ is a variable but $Ds$ is not, we can dynamically identify $Ds$ on the fly (as

we scan the current constraint) as being a parameter which is evidently multiplied with variable *s*, both having index set *i* associated with them. However, we need to pay attention on components appearing iteratively in different or in the same sets of constraints; did we have the component already appearing previously in the model again? In that case we do not have to *declare* it again in the Pyomo model as a new quantity, as that would cause a modeling error.

By *declaration* we mean the real-time execution of a Python command that creates the associated terms inside the Pyomo abstract objected-oriented (OO) model. For instance if a set *i* is identified, the string $model.i = Set(dimen = 1)$ is first written inside the text version of the Pyomo model file, and then on-the-fly executed independently inside the already parsing Python function using the *exec* command. The execution commands run in a sequential manner. All the different possible cases of relationships between parameters and variables are dynamically identified, and the parser keeps track of the local (per constraint) and global (per model) list of parameters identified while scanning the model in dynamically growing lists.

Dynamic identification of the parameters and index sets is one of the elegant features of the platform, since in most Algebraic Modeling Languages (AMLs) the user explicitly defines the model parameters one-by-one. In our case, this is done in an intelligent automated manner. Another important aspect of the decomposition process is the identification of the constraint type ($<=, =, >=$), since the position of the operator is crucial to separate the left and the right hand side of the constraint. This is handled by an independent function. Decomposition also helps identify Quadratic terms. By automatic conversion of the caret symbol to $**$ (as this is one of the ways to denote power of a variable in Pyomo language) the *split* function carefully transfers this information intact to the Pyomo model.

### Summations and conditional indexing

Summation terms need to be enclosed inside parentheses ($\cdots$), even with a single component. This accelerates identification of the summation terms with clarity and consistency. Summations are in a way very different than processing a simplified mathematical expression in the sense that we impose restrictions on how a summation can be used. First of all, the corresponding function to process summations tries to identify how many summation expressions exist in each constraint at a time. Their respective indexing expressions are extracted and then sent back to the index identification functions to be processed. The assignment of conditional indexing with the corresponding summation is carefully managed. Then, the summation commands for the Pyomo model file are gradually built. Summations can be expressed in the following form, and two different fields can be utilized to exploit conditional indexing (upper and lower brackets):

```
\sum\limits_{p: X_{n,p} = 1}^{}(1-sb_{p,k})
```

which then compiles to: $\sum_{p:X_{n,p}=1}(1-sb_{p,k})$

This means that the summation will be executed for all values of *p*, (that is for $p = 1 : |p|$) but only when $X_{n,p} = 1$ at the same time. If we want to use multiple and stacked summations

(double, triple etc.) we can express them in the same way by adding the indexes for which the summation will be generated, as for example:

```
\sum\limits_{i,j}^{}(X_{i,j})
```

which then compiles to: $\sum_{i,j}(X_{i,j})$

and will run for the full cardinality of sets $i, j$. Dynamic (sparse) sets imposed on constraints can be expressed as:

```
X_{i,j} = Y_{i,j} \forall (i,j) \in C \\
```

which then compiles to: $X_{i,j} = Y_{i,j} \quad \forall (i,j) \in C$

This means that the constraint is being generated only for those values of $(i,j)$ which belong to the dynamic set $C$. In order to achieve proper and precise processing of summations and conditional indexing, we have built two separate functions assigned for the respective tasks. Since specific conditional indexing schemes can take place both for the generation of an entire constraint or just simply for a summation inside a constraint, two different sub-functions process this portion of information. This is done using the *\forall* command at the end of each constraint, which changes how the indexes are being generated for the *vertical* expansion of the constraints from a specific index set. Concerning summations it is done with the bottom bracket information for *horizontal* expansion, as we previously saw, for instance, with $p : X_{n,p} = 1$.

A series of challenges arise when processing summations. For instance, which components are inside a summation symbol? A variable that might appear in two different summations at the same constraint can cause confusion. Thus, using a binary list for the full length of variables and parameters present in a constraint we identify the terms which belong to each specific summation. This binary list gets re-initialized for each different summation expression. From the lower bracket of each summation symbol, the parser is expecting to understand the indexes for which the summation is being generated. This is done by either simply stating the indexes in a plain way (for instance $a, b$) or if a more complex expression is used, the for-loop indexes for the summations are found *before* the colon symbol (:).

## Constraint indexing

At the end of each constraint, the parser identifies the " $\forall$ " (*\forall*) symbol which then helps understand for which indexes the constraints are being sequentially generated (vertical expansion). For instance, $\forall (i,j) \in C$ makes sure that the constraint is not generated for all combinations of index sets $i, j$, but only the ones appearing in the sparse set $C$. The sparse sets are being registered also on the fly, if found either inside summation indexing brackets or in the constraint general indexing (after the $\forall$ symbol) by using the keywords *\in*, *\notin*. The simplest form of constraint indexing is for instance:

$$\sum_{j:i \neq j}(x_{i,j}) = 1 \quad \forall i,$$

where the constraint is vertically expanding for all elements of set $i$ and the summation is running for all those values of set $j$ such that $i$ is not equal to $j$. More advanced cases of

constraint conditional indexing are also identified, as long as each expression is separated with the previous one by using a comma. For example in:

$$\forall i < |i|, j \geq i+1$$

we see each different expression separated so the parser can process the corresponding indexing. Three different functions handle identification on constraint- level and the input for the general function that combines these three, accepts as input the whole expression. We process each component (split by commas) iteratively by these three functions:

1. to identify left part (before the operator/reserved keyword/command),
2. the operator and
3. the right-hand part.

For example, in $i < |i|$, the left part is set $i$, the operator is $<$ and the right-hand part is the cardinality of set $i$. In this way, by adding a new operator in the acceptable operators list inside the code, we allow expansion of supported expressions in a straightforward manner.

## AN ILLUSTRATIVE PARSING EXAMPLE

Let us now follow the sequential steps that the parser takes to convert a simple example. Consider the well-known *transportation problem*:

$$\text{minimize} \quad \sum_{i,j}(c_{i,j}x_{i,j})$$

$$\text{subject to:}$$

$$\sum_{j}(x_{i,j}) \leq a_i \quad \forall i$$

$$\sum_{i}(x_{i,j}) \geq b_j \quad \forall j$$

$$x \in \mathbb{R}_+$$

We will now provide in-depth analysis of how each of the main three parts in the model can be processed.

### Variables

The parser first attempts to locate the line of the *.tex* model file that contains the variable symbols and their respective domains. This is done by trying to identify any of the previously presented reserved keywords specifically for this section. The parser reaches the bottom line by identifying the keyword *mathbbR_+* in this case. Commas can separate variables belonging to the same domain, and the corresponding parsing function splits the collections of variables of the same domain and processes them separately.

In this case, the parser identifies the domain and then rewinds back inside the string expression to find the variable symbols. It finds no commas, thus we collect only one variable with the symbol $x$. The platform then builds two Python lists with the name of the variables found and their respective types.

## Objective function

The parser then reads the optimization sense (by locating the objective function expression using the keywords, in this case *minimize*) and tries to identify any involved variables in the objective function. In a different scenario, where not all of the model variables are present in the objective function, a routine identifies one-by-one all the remaining variables and their associated index sets in the block of the given constraint sets.

The parser first attempts to locate any summation symbols. Since this is successful, the contained expression is extracted as $c_{\{i,j\}}x_{\{i,j\}}$, by locating the parentheses bounds (). In case of multiple summations, or multiple expressions inside the parentheses, we process them separately. The bounds of the summation symbol (the lower and upper brackets) respectively will be analyzed separately. In this case, the upper one is empty, so the lower one contains all the indexes for which the summation has to scale. Separated by commas, a simple extraction gives $i, j$ to be used for the Pyomo for-loop in the expression. There is no colon identified inside the lower bracket of the summation, thus no further identification of conditional indexing is required.

A *split* function is then applied on the extracted mathematical expression $c\_\{i,j\}x\_\{i,j\}$ to begin identification of the involved terms. Since there are no operators $(*, +, -, /)$ we have a list containing only one item; the combined expression. It follows that the underscore characters are used to frame the names of the respective components. It is easy to split on these characters and then create a list to store the pairs of the indexes for each component. Thus, a sub-routine detects the case of having more than just one term in the summation-extracted expression. In this example, $c$ is automatically identified as a parameter because of its associated index set which was identified with the underscore character and since it does not belong to the list of variables.

The global list of parameters is then updated by adding $c$, as well as the parameters for the current constraint/objective expression. This helps us clarify which parameters are present in each constraint as well as the set of parameters (unique) for the model thus far, as scanning goes on. Once the parameter $c$ and variable $x$ are identified and registered with their respective index sets, we proceed to read the constraint sets. The parser creates expressions as the ones shown below for this kind of operations:

```
model.i = Set(dimen=1) \\
model.j = Set(dimen=1) \\
model.c = Param(model.i,model.j, initialize = 0) \\
model.x = Var(model.i,model.j, domain=NonNegativeReals) \\
```

Since the objective function summation symbol was correctly identified with the respective indexes, the following code is generated and executed:

```
def obj_expression(model):
  model.F = sum(model.c[i,j]*model.x[i,j] for i in model.i for j in model.j)
  return model.F
model.OBJ = Objective(rule=obj_expression, sense = minimize)
```

## Constraints

Since the constraints sets are very similar, for shortness we will only analyze the first one. The parser first locates the constraint type by finding either of the following operators

$\le, \ge, =$. It then splits the constraint in two parts, left and right across this operator. This is done to carefully identify the position of the constraint type operator for placement into the Pyomo constraint expression later on.

The first component the parser gives is the terms identified raw in the expression ($['x'_{i,j}, 'a'_i]$). Parameter $a$ is identified on the fly and since $x$ is already registered as a variable and the parser proceeds to only register the new parameter by generating the following Pyomo expressions:

```
model.a = Param(model.i, initialize = 0)
```

The platform successfully identifies which terms belong to the summation and which do not and separates them carefully. Eventually the $\forall$ symbol gives the list of indexes for which the constraints are being generated. This portion of information in the structure of a Pyomo constraint definition goes in replacing $X$ in the following piece of code:

```
def axb_constraint_rule_1(model,X):
```

and the full resulting function is:

```
def axb_constraint_rule_1(model,i):
   model.C_1= sum(model.x[i,j] for j in model.j) <= model.a[i]
   return model.C_1
model.AxbConstraint_1=Constraint(model.i,rule=axb_constraint_rule_1)
```

## DISCUSSION

Developing a parser that would be able to *understand* almost every different way of writing mathematical models using LaTeX is nearly impossible; however, even by framing the way the user could write down the models, there are some challenges to overcome. For instance, the naming policy for the variables and parameters. One would assume that these would cause no problems but usually this happens because even in formal modeling languages, the user states the names and the types of every component of the problem. Starting from the sense of the objective function, to the names and the types of the variables and parameters as well as their respective sizes and the names of the index sets, everything is explicitly defined. This is not the case though in this platform; the parser recognizes the parameters and index sets with no prior given information. This in turn imposes trade-offs in the way we write the mathematical notation. For instance, multiple index sets have to be separated by commas as in $x_{i,j}$ instead of writing $x_{ij}$.

On the other hand, using symbolic representation of the models in LaTeX can enable the user quickly identify errors in the description of the model, the involved variables, parameters or their mathematical relationships therein. This as opposed trying to debug models that have been developed directly in a programming language or in an AML, which would make the detection of such errors or typos more challenging.

By scanning a constraint, the parser quickly identifies as mentioned the associated variables. In many cases parameters and variables might have multiple occurrences in the same constraint. This creates a challenging environment to locate the relationships

of the parameters and the variables since they appear in multiple locations inside the string expressions and in different ways. On top of this, the name of a parameter can cause identification problems because it might be a sub/super set of the name of another parameter, e.g., parameter AB, and parameter ABC. Therefore naming conflicts are carefully resolved by the platform by meticulously identifying the exact location and occurrences of each term.

The CPU time required for each step in the modeling process of the platform (conversion from LaTeX to Pyomo, Pyomo model generation, Solver) can be found in the Supplementary Information. It can be noted that the parser is the least consuming step, which clearly demonstrates the efficiency of the platform. The Pyomo model generation and solver (CPLEX in our measurements) steps and their associated CPU-time are completely outside of the parser's control. However, it is essential to get an idea of how these timings compare to each other with the addition of this extra higher level of abstraction in the beginning of the modeling process.

Challenges also arise in locating which of the terms appearing in a constraint belong to summations, and to which summations; especially when items have multiple occurrences inside a constraint, there needs to be a unique identification so as to include a parameter (or a variable) inside a specific summation or not. We addressed this with the previously introduced binary lists. Then, for each of those summation symbols, the items activated (1) are included in the summation or not (0) and the list is generated for each different summation within the expression.

Additionally, another challenge constitutes the extension of the platform to support nonlinear terms, where each term itself can be a combination of various operators and mathematical functions.

Finally, it is worth mentioning that the amount of lines/characters to represent a model in LaTeX in comparison with the equivalent model in Pyomo is substantially smaller. In this respect, the platform accelerates the modeling development process.

## CONCLUSIONS

We presented a platform for rapid model generation using LaTeX as the input language for mathematical programming, starting with the classes of LP, MILP and MIQP. The platform is based on Python and parses the input to Pyomo to successfully solve the underlying optimization problems. It uses a simple GUI to facilitate model and data input based on Django as the web-framework. The user can exploit locally installed solvers or redirect to NEOS server. This prototype platform delivers transparency and clarity, speedup of the model design and development process (by significantly reducing the required characters to type the input models) and abstracts the syntax from programming languages and AMLs. It therefore delivers reproducibility and the ability to replicate scientific work in an effective manner from an audience not necessarily versed in coding. Future work could possibly involve expansion to support nonlinear terms as well as differential and algebraic equations, sanity checking and error catching on input, the ability to embed explanatory comments in the input model file which would transfer to the target AML,

extending the functionality concerning bounds on the variables as well as adding further support to built-in LaTeX commands (such as $\backslash left$[]) which would capture more complex mathematical relationships.

## ACKNOWLEDGEMENTS

We would like to thank Prof. Eric Fraga and Dr. Aristotelis Kittas for useful discussions.

### Funding

This work was supported by The Leverhulme Trust under Grant (No. RPG-2015-240) and the UK Engineering and Physical Sciences Research Council (No. EP/M027856/1). The funders had no role in study design, data collection and analysis, decision to publish, or preparation of the manuscript.

### Grant Disclosures

The following grant information was disclosed by the authors:
Leverhulme Trust under Grant: RPG-2015-240.
UK Engineering and Physical Sciences Research Council: EP/M027856/1.

### Competing Interests

The authors declare there are no competing interests.

### Author Contributions

- Charalampos P. Triantafyllidis performed the experiments, analyzed the data, prepared figures and/or tables, performed the computational work, authored or reviewed drafts of the paper.
- Lazaros G. Papageorgiou conceived and designed the experiments, analyzed the data, performed the computational work, authored or reviewed drafts of the paper, approved the final draft.

### Data Deposition

The raw data used for examples presented in this paper are provided in the Supplemental File.

### Supplemental Information

Supplemental information for this article can be found online at http://dx.doi.org/10.7717/peerj-cs.161#supplemental-information.

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
