# Peer review of "An integrated platform for intuitive mathematical programming modeling using LaTeX"

_PeerJ Computer Science, doi:10.7717/peerj-cs.161_

## Round 0.1 · original submission · Minor Revisions

The two reviewers have made some valid comments and raised a few issues that require your attention. Consequently, I would invite you to revise the paper taking into account their observations.

·

Basic reporting

No comment (see reviewed PDF).

Experimental design

No comment (see reviewed PDF).

Validity of the findings

No comment (see reviewed PDF).

Reviewer 2 ·

Basic reporting

The paper is well-organized, and the use of the English language is generally good.

In line 84, the authors state that “this is the first prototype workable scheme to address how LaTeX could be used as an input language to perform mathematical programming modeling.” This statement is not necessarily true. I encourage the authors to take a closer look at MOSAICmodeling (www.mosaic-modeling.de), which is a modeling and optimization framework based on a LaTeX-style syntax for inputting algebraic and differential equations. A comparison between MOSAICmodeling and the proposed framework would be helpful in terms of evaluating the advantages of this contribution.

In lines 92-97, 22 references are listed with no further details, which is not very useful for the reader. Also, this seems to be a fairly biased list, with contributions mostly from the process systems engineering (PSE) community and many papers co-authored by the authors of this work. I recommend selecting a more representative set of papers from a wider range of fields, considering that the topic addressed here is of interest to more than just the PSE community.

Experimental design

The proposed approach is sound and it makes natural sense to divide the work process of the parser into three parts: objective function, constraints, and variables. However, I have some questions that I hope the authors could answer in their response or address in the revised manuscript:
1. The authors state that the parser can handle Greek letters. How about variables that consist of a Greek and a Latin letter, e.g. \Delta t?
2. How does the parser handle superscripts? Does the parser differentiate between superscripts and exponents?
3. In the literature, summations are often defined over sets instead of explicit conditions, e.g. \sum\limits_{i \in K} with K being a subset of the full-cardinality set I. Can the parser handle this case?
4. Looking at the example in line 347, it seems to be possible to use subsets for constraint generation. How are these subsets defined as these definitions usually do not explicitly appear in the LaTeX formulation of an optimization model? Is this information possibly directly extracted from the .dat file?

Validity of the findings

The validity of the proposed approach is nicely demonstrated in an illustrative example. However, some information on the platform’s ability to detect errors in the LaTeX formulation would be useful. For example, in the illustrative parsing example, what happens if in one of the constraints, the user forgot to write one of the two indices of the variable x?

Moreover, I personally would welcome a discussion on the challenges of extending this platform to the nonlinear case.

---

## Round 0.2 · accepted · Accept

The reviewers are satisfied with your corrections,

·

Basic reporting

No comment.

Experimental design

No comment.

Validity of the findings

No comment.

Additional comments

The revised article, as well as the authors' comments, satisfactorily address my concerns from the first review.

Reviewer 2 ·

Basic reporting

The authors have addressed all the reviewers' comments in the revised manuscript.

Experimental design

The authors have addressed all the reviewers' comments in the revised manuscript.

Validity of the findings

The authors have addressed all the reviewers' comments in the revised manuscript.

Additional comments

The authors have addressed all the reviewers' comments in the revised manuscript.